# Field-level clothianidin exposure affects bumblebees but generally not their pathogens

Dimitry Wintermantel[1,2,3], Barbara Locke [1], Georg K.S. Andersson [4,5], Emilia Semberg[1], Eva Forsgren[1], Julia Osterman[1,6,7], Thorsten Rahbek Pedersen[8], Riccardo Bommarco [1], Henrik G. Smith[4,5], Maj Rundlöf [4,9] & Joachim R. de Miranda [1]

Neonicotinoids are implicated in bee declines and laboratory studies imply that they impair the bee immune system, thereby precipitating a rise in pathogen levels. To establish whether such synergisms reduce bee performance in real-world agricultural landscapes, we analysed the microbial composition of the bumblebee (*Bombus terrestris*) samples from our recent landscape study on the impacts of field-level clothianidin exposure. We related clothianidin exposure and microbial composition to both individual- and colony-level performance parameters, to better understand the direct and indirect mechanistic effects of neonicotinoid exposure on bumblebees. We show that exposure to clothianidin from seed-coated oilseed rape reduces bumblebee size and numbers, particularly of reproductives. However, exposure does not affect the levels of non-pathogenic bacteria or viruses, nor induce rises in the levels or virulence of intracellular parasites. We conclude that field exposure to the neonicotinoid clothianidin affects bumblebee performance but generally not their pathogenic or beneficial microbiota.

[1] Department of Ecology, Swedish University of Agricultural Sciences, 750 07 Uppsala, Sweden. [2] INRA, UE 1255 APIS, Le Magneraud, 17700 Surgères, France. [3] Centre d'Etudes Biologiques de Chizé, UMR 7372, CNRS & Université de La Rochelle, 79360 Villiers-en-Bois, France. [4] Department of Biology, Lund University, 223 62 Lund, Sweden. [5] Centre for Environmental and Climate Research, Lund University, 223 62 Lund, Sweden. [6] Martin Luther University of Halle-Wittenberg, Institute of Biology, 06120 Halle, Germany. [7] Department of Computational Landscape Ecology, Helmholtz Centre for Environmental Research-UFZ Leipzig, ESCALATE, 04318 Leipzig, Germany. [8] Swedish Board of Agriculture, 551 82 Jönköping, Sweden. [9] Department of Entomology and Nematology, University of California, Davis, CA 95616, USA. Correspondence and requests for materials should be addressed to D.W. (email: dywintermantel@gmail.com)

Bees are essential pollinators and their recent declines may have adverse effects on both natural plant biodiversity and the production of crops that depend on insect pollination[1]. Global declines of bees have been attributed to habitat destruction, pesticide use, pathogens, climate change, or some combination of these factors[1,2]. The conversion of often flower-rich natural or semi-natural habitat to arable land is thought to play a major role in the long-term decreases in bee diversity through habitat loss and fragmentation[2]. Moreover, chronic exposure to agricultural pesticides, particularly the neurotoxic neonicotinoids, has recently been implicated in bee declines (reviewed in refs. [2,3]).

Neonicotinoids are used worldwide to control insect pests of economically important crops[4]. They are taken up systemically to all parts of the plant, including pollen and nectar—the major foods for bees. Artificial feeding experiments with neonicotinoid exposure levels that are comparable to residue concentrations found in pollen and nectar of crops and wild flowers[3,5] showed a variety of sublethal effects on bee reproduction[6,7], homing success[8,9], foraging behaviour[9–11], crop pollination[12] and immune function[13], all of which may ultimately cause colony failure[14]. Field studies[15–17] and UK-wide surveys[18,19] confirmed adverse effects of neonicotinoid seed dressings in oilseed rape on bees, but results varied with spatial location[17] and bee species[16] and some studies found no effects[20–22].

To understand the sources of this variation, it is important to identify the mechanisms by which neonicotinoids impact bee performance under field conditions. Mortality in social bees can be masked by colony compensation mechanisms, such as the trade-off between worker production and more energy-costly males[15] or between colony size and individual body size of all bee castes[23]. Regional differences in the effects of field-level neonicotinoid exposure on wild and managed bees have been partly ascribed to regional differences in parasite levels[17], as the impact of neonicotinoids on bees had previously been shown to interact with pathogens and parasites[7,13,24–29] (but see refs. [29–32]). Neonicotinoid exposure was shown to increase pathogen abundance in honeybees[13,24–26] but not in bumblebees[7,32] and to act synergistically with pathogens in increasing mortality of honeybees[25,27,28] and bumblebees[7]. Immune functions in individual bees can be weakened by neonicotinoid exposure, either by suppressing immune genes, which has been shown to stimulate virus replication[13,33], or by impairing individual immunocompetence through a reduction in the number of hemocytes, wound-healing, the antimicrobial activity of the haemolymph and levels of phenoloxidase, an enzyme involved in the melanisation of pathogens[34–36]. Neonicotinoid exposure can also impair hygienic behaviour[37,38] and the production of antiseptic compounds that help preserve food stores[27], both major components of social immunity. In addition, bumblebees exposed to neonicotinoids collect less pollen[9–11], risking undernourishment and consequently weakening the bees' immunocompetence[39,40]. Bumblebees socially transmit distinct microbiota[41], which can enhance the ability to live on suboptimal diets[42,43] and protect against pathogens[41,44,45]. However, it remains unclear whether neonicotinoid exposure affects the gut bacteria that potentially contribute to or alter the bees' immunocompetence.

Most studies examining interactions between neonicotinoids and pathogens in bees have been laboratory-based and focused mainly on honeybees and occasionally on bumblebees[7,32,36,46]. Moreover, there remains a distinct deficit of landscape-scale field studies involving real-world neonicotinoid exposure[24]. We recently demonstrated clear harmful effects of the neonicotinoid clothianidin on wild bees in real agricultural landscapes, using a landscape-scale study design with free-flying bumblebees (*Bombus terrestris*) from colonies placed next to fields that were spring-sown with either clothianidin-treated or insecticide-free oilseed rape (*Brassica napus*) seeds[16]. The study showed that bumblebee colonies at clothianidin-treated sites grew less in weight and produced fewer cocoons than those at non-treated sites. Here, we extend this study by counting adult bees of all castes, measuring the body size of premature and adult bumblebees and analysing the bumblebee microbial composition, including both pathogens and beneficial gut symbionts. We test whether field-level clothianidin exposure affects individual-level or colony-level bumblebee performance and the prevalence (i.e. proportion of infected colonies) and abundance of pathogenic and non-pathogenic microorganisms. We also examine potential interactive effects between neonicotinoid exposure and different microorganisms by testing whether microorganism abundance co-varies differently between treatments with bumblebee performance parameters. For this, we combine the data published in Rundlöf et al. (2015) on colony weight and the number of worker/male cocoons with new data on body size and numbers of bees per caste.

Our results confirm that field-level neonicotinoid exposure impairs reproduction in bumblebee colonies, as shown by fewer queens and males. The negative effects of clothianidin exposure on colony-level performance are supported by negative effects at the individual scale, with clothianidin-exposed colonies producing smaller bees. We find, however, no major effect of the neonicotinoid on pathogens, beneficial gut bacteria or their relationship to the host's performance, suggesting that the mechanisms by which clothianidin affects bumblebees in agricultural landscapes are largely independent of the bumblebee microbiota.

## Results

**Number and size of bee pupae and adults.** Colonies at clothianidin-treated fields had on average 234 (41%) fewer total bees (intact cocoons + adults) than colonies at control fields (LRT, $P < 0.001$), despite a similar number of adult workers in the two groups (LRT, $P = 0.53$, Table 1; Fig. 1). The production of reproductives (queens and males) was markedly reduced in colonies at clothianidin-treated fields, as indicated by 32.8 or 66% fewer adult males (LRT, $P < 0.001$) and 71.1 or 74% fewer queens (intact cocoons + adults; $P < 0.001$; see also Rundlöf et al.[16]).

**Table 1 Bee numbers in relation to clothianidin seed treatment**

| Response | Model | Predictor | Estimate (number) | Estimate (%)[a] | $\chi^2_1$ | P-value[b] |
|---|---|---|---|---|---|---|
| Bees (adults + cocoons) | LMM[c] | Treatment | −234.4 | −40.8 | 11.05 | **<0.001** |
| Adult workers | LMM[c] | Treatment | −24.4 | −12.5 | 0.04 | 0.526 |
| Queens (adults + cocoons) | GLMM[d] | Treatment | −71.1 | −73.5 | 17.52 | **<0.001** |
| Adult males | GLMM[d] | Treatment | −32.8 | −65.7 | 17.63 | **<0.001** |

[a]Effect sizes in % were calculated in reference to the control group
[b]P-values were calculated by likelihood ratio tests with 1 degree of freedom and $P < 0.05$ is highlighted in bold
[c]Linear mixed-effects models (LMM; with normal error distribution)
[d]Generalized linear mixed-effects models (GLMM; with negative binomial error distribution and log link)

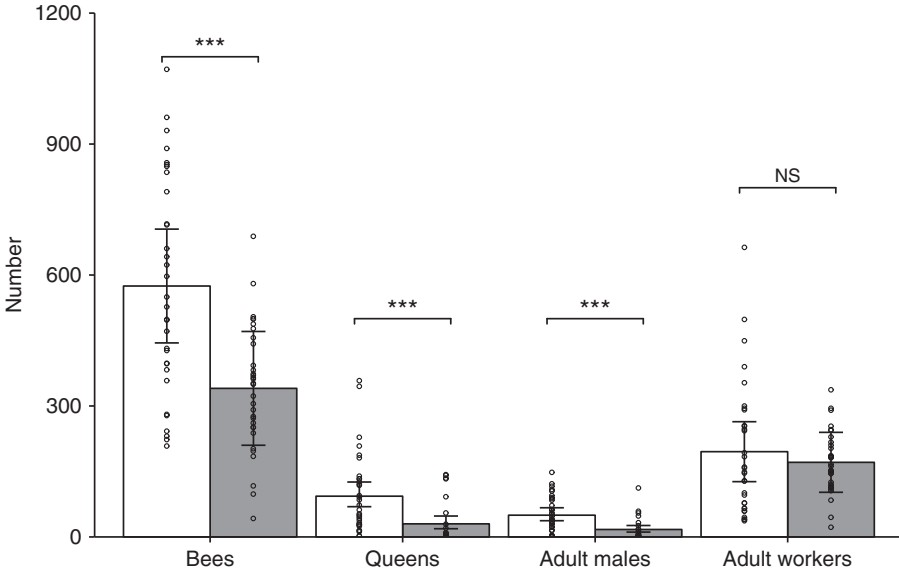

**Fig. 1** Bee numbers. Number of bees (adults + cocoons), queens (adult queens + queen cocoons), adult males and adult workers per bumblebee colony (32 per treatment) in relation to treatment (white, control; grey, clothianidin seed coating) in oilseed rape fields (8 per treatment). The error bars represent 95% profile confidence intervals of linear mixed-effects model estimates. Circles indicate measured values (per colony). NS not significant ($P > 0.05$), ***$P < 0.001$. $P$-values were calculated by likelihood ratio tests on (generalised) linear mixed-effects models

| Table 2 Bee size in relation to clothianidin seed treatment, caste and developmental stage | | | | | | | | | |
|---|---|---|---|---|---|---|---|---|---|
| Sample | Model | Response | Predictor | Estimate | $\chi^2_1$ | $P$-value[a] | $N$ Fields | $N$ Colonies | $N$ Bees |
| All pupae | CLMM[b] | Stage | Caste[d] | −1.32 mg | 34.09 | **<0.001** | 16 | 62 | 678 |
| | LMM[c] | Body mass | Caste[d] | 45.0 mg | 18.52 | **<0.001** | | | |
| Male pupae | CLMM[b] | Stage | Treatment | −0.02 | 0.00 | 0.959 | 16 | 47 | 456 |
| | LMM[c] | Body mass | Treatment | −85.1 mg | 20.40 | **<0.001** | | | |
| | | Stage | | −8.0 mg | 17.10 | **<0.001** | | | |
| Adult workers | LMM[c] | Body mass | Treatment | −21.9 mg | 1.76 | 0.184 | 16 | 64 | 633 |
| | LMM[c] | Intertegular distance | Treatment | −0.26 mm | 5.95 | **0.015** | | | |

[a]$P$-values were calculated by likelihood ratio tests with 1 degree of freedom and $P < 0.05$ is highlighted in bold
[b]CLMM = Cumulative link mixed model (with logistic error distribution and logit link)
[c]LMM = Linear mixed-effects model (with normal error distribution)
[d]Differences between castes are shown in reference to worker pupae

Because, the control colonies tended to be further along in their development than the exposed colonies, we were able to obtain male pupal samples from 28 of 32 colonies at untreated fields, but from only 16 of 32 colonies at clothianidin-treated fields. Similarly, we were able to obtain samples of at least 7 worker pupae more often from clothianidin-exposed (18) than control colonies (4). Samples of both male and worker pupae could be obtained from four clothianidin-exposed colonies that were in the transition from worker to male production, while two exposed colonies had neither worker nor male pupae. Generally, the male pupae were at an earlier developmental stage (LRT, $P < 0.001$, Table 2) and had a larger body mass than the worker pupae (LRT, $P < 0.001$). Only the male pupae data were further analysed, because there were too few control colonies (4) with enough worker pupae to allow meaningful data analyses and interpretation. The male pupae did not differ in developmental stage between treatments (LRT, $P = 0.96$) but were 21.5–23.9% (depending on developmental stage) lighter at clothianidin-treated fields than similar pupae at control fields (LRT, $P < 0.001$; Fig. 2a, Table 2). Overall, the male pupal body mass decreased about 8 mg per developmental stage (corresponding to approximately 2 days; LRT, $P < 0.001$).

The adult workers at clothianidin-treated and control fields had similar body mass (LRT, $P = 0.18$, Table 2; Fig. 2b), but those at treated fields had on average 4.8% (0.26 mm) smaller thoraxes (intertegular distance), than those at control fields ($P = 0.015$).

**Microorganism prevalence**. The two principal symbiotic gut bacteria of adult bees (*Gilliamella apicola* and *Snodgrasella alvi*) were detected at all fields. In each treatment group, *G. apicola* was detected in adult worker bumblebee samples from 91% of the colonies. *Snodgrasella alvi* tended to be more prevalent in colonies at clothianidin-treated fields than in colonies at control fields (LRT, $P = 0.057$, Fig. 3, Supplementary Table 1). The most prevalent pathogen was *Crithidia bombi*, which in contrast to *S. alvi*, was less frequently detected in colonies at clothianidin-treated fields than in control colonies ($P = 0.037$). *Apicystis bombi* was the only other pathogen that was detected in a majority of the colonies, but its prevalence did not differ between treatments (LRT, $P = 0.40$). *Nosema bombi*, *Sacbrood virus* (SBV), *Slow bee paralysis virus* (SBPV) and *Acute bee paralysis virus* (ABPV) were sporadically detected. *Deformed wing virus* type A (DWV-A), *Black queen cell virus* (BQCV), *Chronic bee paralysis virus* (CBPV), *Lake Sinai virus* types 1 and 2 (LSV-1, LSV-2), *Apis mellifera filamentous virus* (AmFV) and the microsporidians

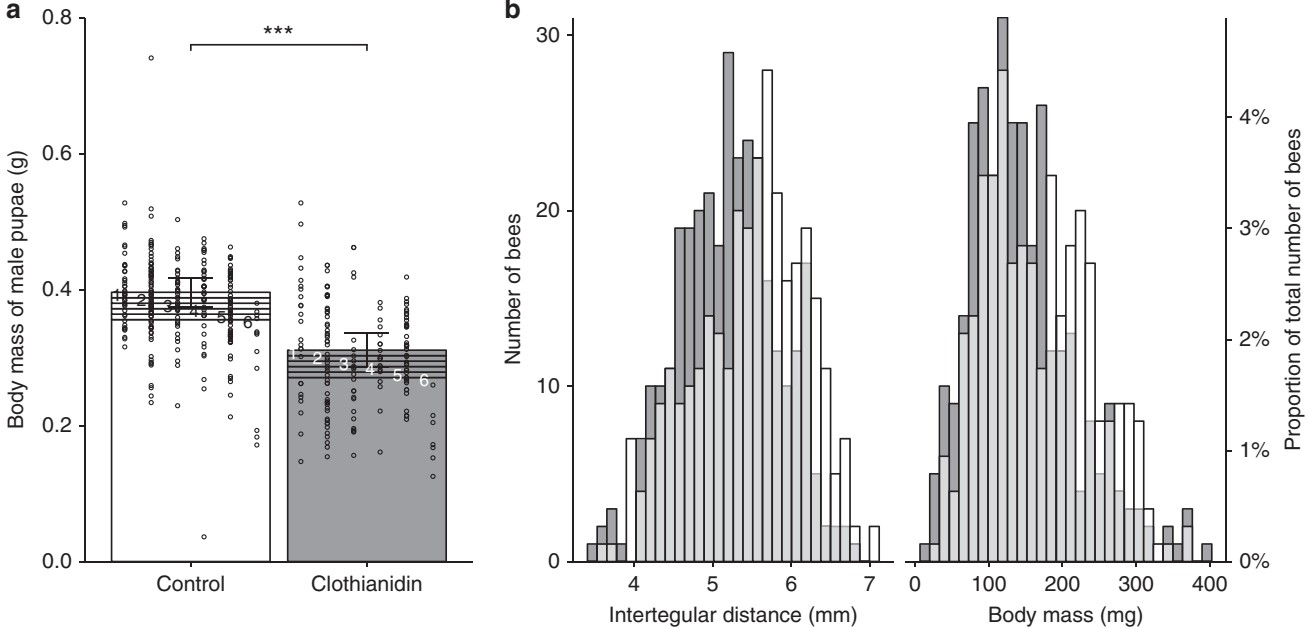

**Fig. 2** Size of pupae and adult bumblebees. **a** Body mass of male pupae in relation to treatment (control or clothianidin seed coating) and pupal developmental stage (1–6). Error bars represent 95% profile confidence intervals of linear mixed-effects model estimates at the earliest developmental stage (1). Circles indicate raw data on measured body mass (per bee). ***$P < 0.001$. $P$-values were calculated by likelihood ratio tests on a linear mixed-effects model. **b** Histograms of the intertegular distance and the body mass of adult worker bumblebees from colonies (32 per treatment) placed in oilseed rape fields (8 per treatment) sown from clothianidin-treated (dark grey; $n = 320$ bees) or insecticide-free seeds (white; $n = 313$). Overlaps between the two treatment groups are shown in light grey and values are expressed in absolute terms and percentages of the total number of measured bees

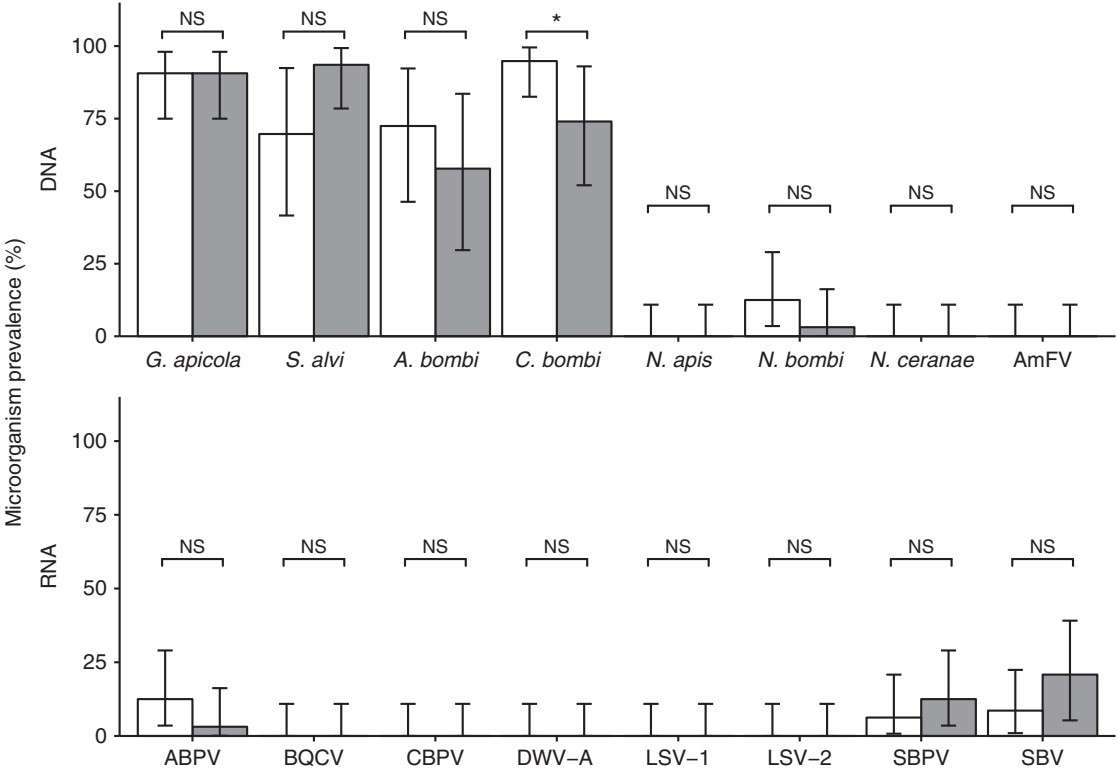

**Fig. 3** Microorganism prevalence. Percentage of bumblebee colonies (32 per treatment) infected with microorganisms in relation to treatment (white, control; grey, clothianidin seed coating) in oilseed rape fields (8 per treatment). For *S. alvi*, *A. bombi*, *C. bombi* and SBV generalized mixed-effects model estimates and their 95% confidence intervals are shown. For all other microbiota controlling for non-independence of colonies placed by the same field was not feasible. Therefore, the actual proportions of infected colonies per treatment and 95% confidence intervals calculated by two-sided binomial tests are illustrated. $P$-values were calculated based on likelihood ratio tests for model estimates and two-sided tests of equal proportions for measured proportions. NS not significant ($P > 0.05$), *$P < 0.05$

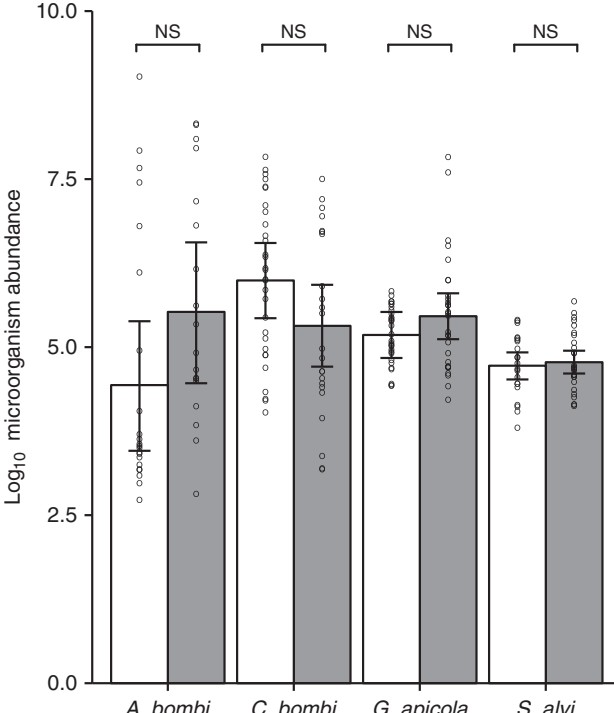

**Fig. 4** Microorganism abundance. The log$_{10}$ DNA copy numbers per bee and colony of the most frequently detected microorganisms (*Apicystis bombi* (n = 40 colonies in N = 15 fields), *Crithidia bombi* (n = 53, N = 16), *Gilliamella apicola* (n = 58, N = 16), *Snodgrassella alvi* (n = 50, N = 16)) in relation to treatment (white, control; grey, clothianidin seed coating) in oilseed rape fields (8 per treatment). The error bars represent 95% profile confidence intervals of linear mixed-effects model estimates. Circles indicate measured values (per colony). NS not significant (P > 0.05). P-values were calculated by likelihood ratio tests on linear mixed-effects models

*Nosema apis* and *Nosema ceranae* were not detected in any of the 64 bumblebee colonies. With the exception of two ABPV-infected control colonies, the pupal samples were free of viruses. Of the five colonies with ABPV-infected adults, two had also ABPV-infected pupae.

**Microorganism abundance**. Quantitative analyses of microorganism abundance were restricted to the positive samples of the four most prevalent microorganisms (*G. apicola*, *C. bombi*, *S. alvi*, *A. bombi*), in order to test the effects of clothianidin exposure on microorganism abundance independently of any effects on prevalence. Datasets from low-prevalence microorganisms were not analysed, as these were either too small (if non-detections are excluded) or too distorted by excess zero values (if non-detections were included) for meaningful analysis. Clothianidin exposure did not affect the abundance of any of these microorganisms in the infected bumblebee colonies (LRT, P ≥ 0.1, Supplementary Table 2; Fig. 4). Neither did the abundance of any of these microorganisms co-vary with the abundance of any of the other microorganisms (Supplementary Data 1).

**Interaction between clothianidin exposure and microorganisms**. Here, we assessed the interactive effect of clothianidin exposure and the abundance of the four most frequently detected microorganisms on the body size and the numbers of bees (of different castes) as well as on the previously in Rundlöf et al.[16] reported colony weight and number of worker/male cocoons.

Clothianidin exposure affected how *G. apicola* abundance in adult workers co-varied with the body mass of adult workers (P = 0.006) and the number of adult males (P = 0.027, Supplementary Data 2). In the clothianidin-exposed colonies, increase in *G. apicola* abundance was associated with an increase in worker body mass (LRT, P < 0.001; ±47 mg between mean-max log$_{10}$ *G. apicola* abundance, n = 32 colonies) and in the number of adult males (LRT, P = 0.001; ±28.2 bees between mean-max log$_{10}$ *G. apicola* abundance), whereas no such co-variance was found for the control colonies (LRT, worker body mass: P = 0.97; number of adult males: P = 0.92).

No other interactions between treatment and microorganism abundance could be identified, although there was a weak indication that the relationship between the number of adult workers and *A. bombi* abundance differed between treatments (LRT, P = 0.057; Supplementary Data 2).

Independently of treatment, the number of adult males declined with *S. alvi* abundance in adult workers (LRT, P = 0.022; ± 6.9 bees between mean-maximum log$_{10}$ *S. alvi* abundance, n = 64 colonies). We observed no other treatment-independent co-variation between microorganism abundance and bee performance parameters except for a non-significant tendency of the number of worker/male cocoons to decline with the abundance of *A. bombi* in adult worker bees (LRT, P = 0.054).

## Discussion

In this study, we confirm the previously reported negative effects of field exposure to clothianidin seed-treated oilseed rape on bumblebees (*B. terrestris*) at the colony level[16] and show that this is also reflected at the individual bumblebee level, by a reduction in body size. In addition, we provide further evidence for reduced production of reproductives[6,16,17] in colonies next to clothianidin-treated fields compared to colonies next to control fields and find that these adverse effects are unrelated to differences in microbiome composition. The shift from the production of workers to reproductives, which typically occurs after a colony growth phase, seemed to be delayed in colonies at clothianidin-treated fields. Colonies at treated fields had about 70% fewer queens and adult males compared to colonies at control fields, although the number of adult worker bees was comparable between clothianidin-exposed and control colonies. Although, the number of workers may have been affected by the proportion of bumblebees that were not in their nest when these were removed, clothianidin-exposed colonies exhibited also a several-times lower ratio of males to workers among the examined pupae than the control colonies. This reduced, or delayed, production of reproductives may be due to colony compensation mechanisms for worker losses[15], impaired ovary development[32,47] and/or undernourishment due to reduced pollen foraging or insufficient brood care[9–11]. When the colonies were terminated, the developmental stage of the worker brood was clearly more advanced than that of the male brood, suggesting that the colonies were switching from worker production to male/queen production. This suggests that the differential between exposed and control colonies in the preponderance of worker or male brood may represent a delay in, rather than an abandonment of, the production of reproductives.

We observed that colonies at clothianidin-treated fields had not only fewer but also smaller bees. Lighter male pupae and smaller worker bees suggest that clothianidin exposure may directly or indirectly interfere with the development of individual bees. The size of bumblebees as adults is strongly influenced by the amount of food provisioned at the larval stage[48], suggesting that pupae in clothianidin-exposed colonies may have been undernourished as larvae, as a result of impaired pollen foraging success or deficient brood care by the adult bees[9–11]. Such reduced brood care may

reflect a quantitative shift by the colony towards foraging to compensate for a perceived deficit in foraging success, either through insufficient food intake or neuronal changes[11]. *Bombus terrestris* and honeybee workers can be attracted to neonicotinoid contaminated food, even though its consumption can reduce the bees' overall food intake[49]. However, this effect was not observed for clothianidin[49]. As smaller bumblebees are less efficient foragers[50,51], decreased worker size may exacerbate undernourishment, although our previous results showed that clothianidin exposure did not affect the number of food storage cells[16]. Larger body size is also thought to be advantageous for the mating ability of males[52,53], therefore the drastic reduction in body mass of male pupae may have implications for their future reproductive success.

The adverse effects on body size and the production of reproductives are likely not due to increased pathogen susceptibility. Clothianidin-exposed and control colonies showed only little difference in the prevalence or the abundance of symbiotic and pathogenic microbiota. Of the eight RNA viruses tested, only three were detected in the adult bee samples, and furthermore only in a minority of colonies, while only one (ABPV) was detected in pupal samples. Of the six DNA pathogens tested, only the three bumblebee-specific parasites (*A. bombi, C. bombi, N. bombi*) were detected. The prevalences of these pathogens were generally comparable to other studies on commercially reared bumblebees[54,55], although *C. bombi* was detected more frequently in our study. This parasite showed a higher prevalence in the control than in the clothianidin-exposed group, even though previous laboratory-based research could not detect an effect of neonicotinoid exposure on *C. bombi*[7,32]. This may be due to reduced *C. bombi* proliferation, which then leads to lower transmission or detection rates of the parasite. Clothianidin exposure may directly reduce *C. bombi* proliferation or indirectly by inducing changes in bumblebee size demographics, nourishment and foraging activity that also affect *C. bombi* proliferation[56]. Beta-proteobacteria in the bee gut have previously been suggested to protect bumblebees against *C. bombi* infection[41,45] and in our study *S. alvi* tended to be, contrary to *C. bombi*, more prevalent in colonies at clothianidin-treated fields. However, we did not find a relationship between the abundances of *S. alvi* and *C. bombi*. As those bees that failed to return to their nests or were removed by their nest mates after dying could not be sampled, it is conceivable that a decreased survival rate or homing success of *C. bombi* infected bees that lacked *S. alvi* or were additionally immune-challenged by the neonicotinoid[8] caused a lower detection rate of *C. bombi* and masked a negative relation between *C. bombi* and *S. alvi*.

The absence of interactive effects between clothianidin exposure and the abundance of intracellular parasites suggests that clothianidin did not affect the virulence of the parasites or the tolerance of the host to parasite infection. *Gilliamella apicola* was the only microorganism whose relationship with the bumblebee performance parameters depended on clothianidin exposure. In clothianidin-exposed colonies, the body mass of adult worker bees and the number of adult males increased with the abundance of *G. apicola* in adult workers, whereas no co-variation was observed in the control group. *Gilliamella* spp. promote weight gain in bees through the decomposition of otherwise indigestible or toxic carbohydrates[42,43], which may be one possible explanation as to why adult worker bees of the two treatment groups did not significantly differ in body mass, but did differ in intertegular distance. In contrast, to *G. apicola*, *S. alvi* abundance in adult workers showed a negative co-variation with the number of adult males independently of treatment. We are aware that caution has to be applied when interpreting marginally significant effects (such as the effects of microorganisms in interaction or

independently of treatment on the number of adult males or the treatment effect on *C. bombi* prevalence), since the probability of a false-positive finding increases with the number of parameters tested and can be much greater than the probability of falsely rejecting the null hypothesis evaluated by *P*-values[57].

The exposure of bumblebees to clothianidin from seed-coated oilseed rape during one flowering season, which was confirmed by residue analysis of bumblebee-collected nectar[16], did not affect the levels of symbiotic gut bacteria or viruses, nor did it induce rises in the levels or virulence of intracellular parasites. This suggests that the bumblebees' combined individual, adaptive and social immune defences[7,13,33–36] were not sufficiently affected to impair colony-level pathogen susceptibility during this time interval, and that the mechanisms by which exposure to clothianidin affects bumblebee colonies are largely independent from those affected by biological pathogens or diseases. Two comparable field-level studies, but with free-foraging honeybees instead of bumblebees (and lacking adequate site replication), obtained contrasting results for the effects of neonicotinoids on parasite and pathogen levels in honeybees. One study found no impact of clothianidin on *Varroa* and virus levels[58], while the other detected in the first year of the experiment an increase in physiological stress, BQCV and *Varroa* abundance in honeybee colonies placed by neonicotinoid-treated maize fields relative to colonies at untreated maize fields, even though neonicotinoid exposure could not be confirmed[23]. The surviving honeybee colonies were placed for an additional season by the maize fields of the same treatment (control/clothianidin or thiamethoxam) and *Varroa* abundance was again higher in colonies placed by neonicotinoid-treated fields with low levels of neonicotinoid exposure confirmed for apiaries at treated sites and for one of two control apiaries[59]. Experimentally induced exposure of honeybees to both *Varroa* and clothianidin spiked syrup showed no interactive effect between the two pressures[31]. In another study, long-lasting in-hive feeding of thiacloprid to honeybee colonies did not affect colony performance or the levels of parasites, pathogens and expressed immunity-related genes[30]. Thiacloprid belongs, however, to the group of cyano-substituted neonicotinoids, which are substantially less toxic to bees than nitro-substituted neonicotinoids, such as clothianidin or imidacloprid[3]. Pettis et al. (2012)[26] conducted two trials with emerging workers taken from colonies fed with imidacloprid to investigate synergism between the neonicotinoid and *Nosema* spp. They found that imidacloprid increased spore counts when the pathogens were administered with food, but decreased spore counts when *Nosema* was naturally acquired. Pathogen-pesticide interaction in bumblebees has been studied under laboratory conditions with the pyrethroid λ-cyhalothrin and *C. bombi*[60]. Chronic exposure to the pyrethroid did not affect *C. bombi* prevalence or abundance but the body mass of *B. terrestris* workers. Other individual-level or colony-level performance parameters were unaffected by the treatment[60].

Neonicotinoids and certain bee pathogens (predominantly viruses) share a common target: the bee's nervous system[61]. This is a plausible causative explanation for the synergism observed between neonicotinoids and pathogens in laboratory studies, particularly at high levels of infection and pesticide exposure[13,25,35]. In the field, however, pesticide–pathogen synergism may be masked by potentially more potent drivers of pathogen prevalence and abundance, such as population dynamics of bee communities, nutrient availability or adapted foraging behaviour[56,62].

We conclude that exposure to clothianidin seed-treated oilseed rape impacts bumblebees at both the colony and the individual level but does not increase their susceptibility to pathogens. The strong effects of clothianidin exposure on the production of

queens and males suggest that neonicotinoids may deleteriously influence bumblebee population sizes[19], which may further be exacerbated by a neonicotinoid-induced reduction in colony initiation after hibernation[32]. Long-term studies at the population level are needed to investigate whether bumblebee colonies can recover from temporarily reduced brood production during periods when they are no longer exposed to neonicotinoids, as shown under laboratory conditions[63].

## Methods

**Study sites**. In 2013, a total of 16 fields (field size = 8.9 ± 1.4 ha (mean ± s.e.m)) in southern Sweden, intended for the production of spring-sown oilseed rape (*Brassica napus* L.), were selected by the absence of other oilseed rape fields within a 2 km radius and paired based on geographical proximity and the land use in the surrounding landscapes (*r* = 2 km; for details see Rundlöf et al.[17]). For each field pair, one field was randomly assigned to be sown with clothianidin-treated oilseed rape seeds (25 mL Elado (Bayer; 400 g L$^{-1}$ clothianidin + 180 g L$^{-1}$ b-cyfluthrin) per kg seed and the fungicide thiram) while the paired field was sown with insecticide-free oilseed rape seeds treated only with thiram. Farmers at both treated and untreated fields used non-neonicotinoid insecticide sprays but were instructed not to use other neonicotinoids for pest control (see Rundlöf et al.[17]). However, one control field was accidently sprayed with 0.3 L ha$^{-1}$ Biscaya, which contains the neonicotinoid thiacloprid as active ingredient. Thiacloprid has a considerably lower acute toxicity to bees than clothianidin[3]. Residue analyses of bee-collected pollen and nectar revealed that the overwhelming insecticide exposure, and greatest differential between control and treated fields, was from clothianidin, with minor traces of the spray insecticides distributed equitably between control and treated fields[16].

**Bumblebee colonies**. Six commercially reared *Bombus terrestris* colonies (Naturpol beehives, Koppert Biological Systems) were placed in triplets in two wooden, ventilated houses placed in shaded areas along the field edge in each of the 16 fields between June 14 and 28, 2013, at the onset of oilseed rape flowering in each field (for details see Rundlöf et al.[16]). Field allocation was randomized and there was no difference in weights between colonies at treated (723 ± 19 g (mean ± s.e.m.), *n* = 32 colonies) and control (733 ± 18 g, *n* = 32) fields at placement[17]. The colonies were ~10 weeks old at the time of placement, containing roughly 50 workers, one queen, and both pupae and larvae. All 12 bumblebee colonies at a field pair were freeze-killed simultaneously at −20 °C between July 7$^{th}$ and August 5$^{th}$ 2013, at first sighting of new queens in any one of those 12 colonies, one pair after closing nests with check valves for >24 h and the rest at night when most bees were assumed to be in the nest. Since oilseed rape flower phenology influenced placement time and the switch to queen production determined termination time, the duration of field placement varied between 23 and 38 days for different sets of colonies.

**Bee performance parameters**. The two outer bumblebee colonies in each housing unit (total of four from each field) had been assessed previously for the number and weight of queen cocoons, the number and weight of worker/male cocoons, the number of pollen and nectar cells, and the weight of the nest structure[16]. These samples were stored frozen at −20 °C. In this study, we determined the following additional parameters for these same colonies: the total number of bees (adults and pupae), the total number of queens (adults and pupae), the number of adult workers and the number of adult males, as well as the caste, weight and developmental stage of individual pupal cocoons, and the body mass and intertegular distance of individual adult workers. We were unable to categorize ~0.7% of the adult bees, which were not used in analyses. Intertegular distance is the distance between the insertion points of the wings[64] and a standard measure of adult body size in bees. Intertegular distance was measured using a digital caliper and individual body mass of adults and pupae was measured using a balance with 0.1 mg resolution. Only intact bees were analysed. The developmental stage of individual pupae was rated into 6 categories based on eye colour (white = 1, pink = 2, brown = 3), body colour (white = 1–3, brown = 4, black = 5) and the presence of wings (6).

**Pathogens and beneficial bacteria**. The colonies that were assessed for bee performance parameters were also examined for the presence and abundance of the most common and important pathogens and beneficial microbes, including the RNA viruses *Deformed wing virus type A* (DWV-A), *Acute bee paralysis virus* (ABPV), *Black queen cell virus* (BQCV), *Sacbrood virus* (SBV), *Slow bee paralysis virus* (SBPV), *Chronic bee paralysis virus* (CBPV), *Lake Sinai virus* types 1 and 2 (LSV1 & LSV2), the DNA virus *Apis mellifera filamentous virus* (AmFV), the microsporidian gut parasites *Nosema apis*, *N. ceranae* and *N. bombi*, two other common internal parasites (*Apicystis bombi*, *Crithidia bombi*) and the non-pathogenic gut bacteria *Gilliamella apicola* and *Snodgrassella alvi* (Supplementary Table 3).

**Sample processing and homogenization**. For each colony, pooled samples of ten adult worker bumblebees, ten worker pupae and/or ten male pupae were prepared. Two bumblebee colonies (both from treated fields) did not contain any pupae and not all of the remaining colonies contained both worker and male pupae, so that the final sample set consisted of 64 adult samples, 22 worker pupae samples and 44 male pupae samples. Furthermore, five of the worker pupae samples contained only 7–9 individuals.

The 10 bees in each pooled sample were placed in a polyethylene bag with an inner mesh (BioReba). The bees were finely ground with a pestle. One millilitre of nuclease-free water per bee was added and the slurry was mixed thoroughly until the suspension was homogenous. The homogenates were stored at −80 °C in 1 mL aliquots until nucleic acid extraction.

**Nucleic acid extraction**. DNA was extracted from the adult bumblebee homogenates using the protocol for extracting DNA from *Nosema* spores[65], which is sufficiently robust to also extract DNA from bacteria and other microorganisms. One millilitre of primary bee homogenate was centrifuged for 5 min in a microfuge at 13,000 rpm. The pellet was repeatedly frozen-thawed with liquid nitrogen and ground with a sterile teflon micro-pestle until pulverized. The pulverized pellet was re-suspended in 400 µL Qiagen Plant tissues DNeasy AP1 lysis buffer containing 4 µL RNASe-A (10 mg mL$^{-1}$) and incubated and shaken for 10 min at 65 °C, after which 130 µL P3 neutralization buffer (3.0 M potassium acetate pH 5.5) was added, followed by 5 min incubation on ice and centrifugation for 5 min at 14,000 rpm to remove the lysis debris. DNA was purified from 500 µL of the supernatant by the Qiagen automated Qiacube extraction robot, following the plant DNeasy protocol and eluting the DNA into 100 µL nuclease-free water. RNA was extracted by the Qiacube robot directly from 100 µL of both the adult and pupal bumblebee homogenates, using the Qiagen Plant RNeasy protocol (including the Qia-shredder for additional homogenization[66]), eluting the RNA into 50 µL nuclease-free water. The approximate nucleic acid concentration was determined by NanoDrop, after which the samples were diluted with nuclease-free water to a uniform 2 ng µL$^{-1}$ (DNA) or 5 ng µL$^{-1}$ (RNA) and stored at −80 °C.

**RT-qPCR and qPCR**. The RNA pathogens were quantified by Reverse Transcription-quantitative PCR (RT-qPCR). Two technical assays were also included in the RNA analyses: one for RNA250, a passive exogenous reference RNA of known concentration included in the RT-qPCR reaction mixture, for correcting sample-specific differences in assay performance[67], and one for the mRNA of the bumblebee internal reference gene Bt-RPL23[68], for correcting sample-specific differences in RNA quality[69]. The DNA pathogens were quantified by qPCR. Novel qPCR assays were designed, experimentally optimized and confirmed through bidirectional Sanger sequencing of representative PCR products for *Crithidia bombi* (based on the GADH gene), *Nosema ceranae* and *N. bombi* (based on the small subunit ribosomal RNA gene), *Snodgrassella alvi* and *Gilliamella apicola* (based on the 16S ribosomal RNA gene). All sequences matched 100% their intended target.

The PCR reactions were run in duplicate and conducted in 20 µL volumes containing 2 µL template, 0.2 µM (RNA) or 0.4 µM (DNA) of forward and reverse primer (Supplementary Table 3) and either the Bio-Rad EvaGreen qPCR mix (DNA) or the Bio-Rad iScript One-Step RT-qPCR mix (RNA), both with SYBR Green detection chemistry. The reactions were incubated in 96-well optical qPCR plates in the Bio-Rad CFX connect thermocycler, using the following amplification cycling profiles for the RNA assays: 10 min at 50 °C for cDNA synthesis (RT-qPCR only), 5 min at 95 °C (to inactivate the reverse transcriptase and activate the Taq polymerase) followed by 40 cycles of 10 s at 95 °C for denaturation and 10 s at 58 °C for primer annealing, extension, and data collection. For the DNA assays the following amplification cycling profiles were used: 2 min at 98 °C for the initial denaturation followed by 40 cycles of 5 s at 98 °C for denaturation and 10 s at 60 °C for primer annealing, extension, and data collection. The amplification cycles were followed by a melting curve analysis to determine the specificity of the amplification by holding the temperature for 10 s at 95 °C and then reading the fluorescence at 0.5 °C increments from 65 °C to 95 °C. Included on each reaction plate were positive and negative (template-free) assay controls. For each type of assay (Supplementary Table 3) a calibration curve was prepared through a tenfold dilution series of a positive control of known concentration covering seven to eight orders of magnitude, for quantitative data conversion, establishing the reference melting curve profile of the amplicon and estimating the reaction performance statistics.

**Data conversion and normalization**. The melting curves of individual reactions were evaluated visually in order to separate out non-specific amplifications, which differ in melting temperature profiles from true target cDNA/DNA amplicons. Non-specific amplifications were deleted from the data set. All assays were run in duplicate, with the mean value of these two duplicates used in further calculations. Both duplicates had to yield a positive quantitative value and pass the melting curve analysis for the data to be included in the data set. The Cq-values (quantification cycle) of all confirmed amplifications were subsequently converted to estimated SQ-values (Starting Quantity) in copy numbers of each target DNA/RNA, using the corresponding calibration curves for the different assays. These data were

multiplied by the various dilution/conversion factors incurred throughout the extraction, cDNA synthesis and amplification procedures, to calculate the estimated copies of each target per bee The data for the RNA targets were adjusted with the sample-specific data for the two technical assays: the exogenous (RNA250) and internal (Bt-RPL23) reference RNAs. The data for RNA250 was used to calculate the individual cDNA conversion efficiency for each sample, i.e. the ratio of the amount of RNA250 estimated by RT-qPCR (output) versus the known amount of RNA250 added to the reaction (input). As RNA is easily degraded there is a risk that differences between individual samples in RNA quality (i.e. degradation) can affect the results[69]. The data for the RNA targets of interest were therefore normalized to the average value for Bt-RPL23 mRNA, thus correcting the data for sample-specific differences in RNA quality with respect to RT-qPCR performance.

**Statistical analyses**. The data consisted of a range of quantitative biological and pathological parameters, which were measured at colony level (bee numbers, microbiome composition) or at individual level (body mass and intertegular distance). The microbial data were analysed on both their binary (presence/absence) and quantitative (abundance) characters. As no samples were taken prior to exposure, the statistical analyses consisted largely of straight comparisons between colonies at clothianidin-treated fields and control fields. All analyses were done both including and excluding the field where Biscaya (containing the neonicotinoid thiacloprid) was sprayed, to determine whether this influenced the results. Except in two cases, excluding the field sprayed with Biscaya from the data analysis did not move the $P$-values from below to above the 0.05 threshold or vice versa (Supplementary Table 4).

The impact of treatment on the total numbers of bees (cocoons + adults) and the number of adult workers was assessed using linear mixed-effects models (LMM; with normal error distributions), while treatment effects on the total numbers of queens (cocoons + adults) and the number of adult males were analysed using generalized linear-mixed-effects models (GLMM) with negative binomial error distributions and log links. All models on bee numbers contained treatment as a fixed factor and field pair identity and field identity as random factors.

To test whether the developmental stage of male pupae differed from worker pupae, a cumulative link mixed model (CLMM; with logistic error distribution and logit link) was used. Differences in body mass between the two castes were examined using LMMs. All (C)LMMs for data on individual bees contained the field pair identity, field identity and colony identity as nested random effects. Treatment effects on pupae were only tested for males, due to the low incidence of worker pupae in control colonies. To identify whether the developmental stage of male pupae differed between treatments, a CLMM was used. Effects of treatment and developmental stage on male pupal body mass were examined with LMMs that contained both predictors as fixed factors. Body mass estimates for each stage and treatment were predicted, while setting random effects to zero. Effects of clothianidin exposure on the body mass and intertegular distance of adult workers were analysed using LMMs with treatment as a fixed factor.

To test whether or not clothianidin exposure affected microorganism prevalence generalized linear-mixed-effects models with binomial error distribution and logit link with treatment as a fixed factor and field identity as a random factor were used for *A. bombi*, *C. bombi*, *S. alvi* and SBV. For all other microorganisms, the effective sample size (i.e. the less frequent outcome of the presence/absence variable) was too small for the use of random effects. Therefore two-sided tests of equal proportions were conducted using the prop.test function in R to test whether prevalences differed between treatments. Hereby, a Yates' continuity correction was applied to avoid overestimation of statistical significance in small datasets.

Only four microorganisms (*A. bombi*, *C. bombi*, *G. apicola* and *S. alvi*) were prevalent enough for meaningful analysis of their quantitative levels. Microorganism abundances were logarithmically ($log_{10}$) transformed, because microorganism titres are exponentially distributed as a result of their growth dynamics. The impact of clothianidin on microorganism abundance was only tested among colonies infected with the microorganism to test effects on abundance independently of differences in prevalence and to avoid data distributions that are skewed by zero values. LMMs with normal error distributions containing treatment as a fixed effect and field pair identity and field identity as random effects were used to examine whether clothianidin impacted the levels of each microorganism in infected colonies. The co-variation between the abundance of the four most prevalent microorganisms was tested on the data of all colonies, including those that were not infected by one or more of these microorganisms. LMMs predicting the abundance of a microorganism by the abundance of another were conducted for the whole data set as well as separately for each treatment group.

The relation between bee performance parameters and the abundance of the four most prevalent microorganisms was analysed for all 64 colonies. Bee performance parameters included in addition to the bee size and bee number parameters from this study, colony weight and the number of worker/male cocoons reported in Rundlöf et al.[17]. The latter two parameters were analysed using LMMs with a normal error distribution and the other parameters were analysed with (G)LMMs with the same error distribution and link function as described above. (G)LMMs with an interaction term of treatment and the abundance of one of the four

prevalent microorganisms were used to examine whether bee performance parameters were differently related to microorganism abundance in the two treatment groups. These (G)LMMs contained field pair identity and field identity as random factors for colony-level parameters and additionally colony identity for individual-level parameters. The treatment × microorganism abundance interaction was removed from the model if $P > 0.05$. In contrast, if $P < 0.05$, the relation between bee performance parameter and microorganism abundance was examined with separate (G)LMMs for each treatment group. (G)LMMs for only one treatment group contained field identity as random factor for colony-level parameters and field identity and colony identity for individual-level parameters.

Throughout this study, we calculated $P$-values of model estimates by likelihood ratio tests (LRT), because they, in contrast to conventional Wald tests, make no assumptions about the likelihood surface/curve and are therefore considered more reliable[70]. All analyses were conducted using the R Version 3.3.4. The lmer and glmer functions of the R package lme4 were used for LMMs and GLMMs with binomial error distribution, respectively, whereas the glmmTMB function of the R package glmmTMB was used for GLMMs with negative binomial error distribution and the clmm function of the ordinal package was used for CLMMs.

**Power analysis**. We performed a power analysis for treatment effects or interactive effects between microorganism abundance and treatment where $0.05 > P < 0.1$, to assess the effect size we could potentially detect given our study design, replication and model choice. Power was determined for a range of effect sizes at a nominal confidence level of $\alpha = 0.05$ by 1000 Monte Carlo simulations per effect size using the powerSim function of the simr package in R. For graphical illustration, effect sizes were transformed to percentages. For interactive effects between treatment and microorganism abundance on bee performance, effect size of a $log_{10}$ unit increase in microorganism abundance was shown relative to the estimated response value of a control colony with average microorganism $log_{10}$ abundance. For microorganism prevalence, effect size was shown as a percentage change of infected colonies relative to the total number of colonies.

The power analysis suggested that 80% power was reached for effect sizes ranging from under 15% (number of adult workers by *A. bombi* × treatment) to over 25% (*S. alvi* prevalence by treatment) (Supplementary Fig. 1).

## Data availability
The data supporting the findings of this study are available within the paper, its supplementary information files and/or Rundlöf et al.[16]. The datasets generated and/or analysed during the current study are available from the corresponding author on reasonable request.

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

## Acknowledgements

We thank the farmers for collaboration, the project group for feedback, M. Boch, A. Anderson, C. Du Rietz, A. Jönsson and B. Klatt for weighing and examining bumblebee colonies, and B. Bolker and M. Andersson for statistical advice. The project was funded by the Swedish Civil Contingencies Agency (R.B., T.R.P., H.G.S.), the Carl Tryggers Foundation for Scientific Research, the Royal Physiographic Society, the Swedish Research Council grant no. 330-2014-6439 (M.R.), and FORMAS (H.G.S., R.B.). Analyses of the bumblebee microbiota were financed internally at SLU.

## Author contributions

Conceptualization, J.R.d.M., M.R., R.B., H.G.S., T.R.P., J.O., D.W.; Methodology, J.R.d.M., D.W., E.F., E.S., M.R., G.K.S.A.; Investigation, D.W., E.S., G.K.S.A.; Formal analysis, D.W., M.R.; Data curation, J.R.d.M., M.R.; Writing, D.W., B.L.; Editing, All; Visualization, D.W.; Supervision J.R.d.M., M.R.; Project Administration, E.F., T.R.P., R.B., M.R., H.G.S.; Funding Acquisition, J.R.d.M., M.R., R.B., H.G.S., T.R.P.

## Additional information

**Competing interests:** The authors declare no competing interests.

