## [Peer Review File · Nature Communications]

Reviewers' comments:

Reviewer #1 (Remarks to the Author):

Key results: This study is a companion paper to Rundlöf et al., 2015 which found that OSR seed treatment-derived clothianidin negatively affected bumblebee (*Bombus terrestris*) colony production of reproductives (queens and males) and colony productivity. This second manuscript of the same field experiment by Wintermantel and coauthors addressed the rather important question of whether the observed clothianidin effects on bumblebee colonies were attributable to deleterious effects on bumblebee microbiota (pathogens and gut bacteria). This is the first study to examine interactions between neonicotinoids and bumblebee microbes in an agricultural setting. Rather convincingly, this study found little to no effects of OSR clothianidin under field conditions on pathogens and, for the most part, gut microbes. The authors' examination of colony performance-microbe interactions provides evidence that the cause of clothianidin's impact on bumblebee colonies lies elsewhere. This experiment was especially noteworthy for its robust field design – 96 colonies were examined across a forage period at 16 OSR treatment plots (one clothianidin, one untreated) paired at 8 field sites – exactly the kind of realistic field study that is needed (see Collison et al., 2016, *Biological Reviews*). The authors could also provide additional information (see Suggested Improvements) that would sharply detail the underlying specific impacts of clothianidin on reproductive and worker brood production that affects bumblebees so severely.

Validity: The manuscript contains no uncorrectable flaws that should prohibit its publication. One disagreement between the text and figures needs to be addressed, but it is relatively minor.

Originality and Significance: This manuscript (combined with its companion publication Rundlöf et al., 2015) is original in that it is the first to examine the effects of neonicotinoids on bumblebee microbes under realistic agricultural exposures in the field. All and all, this study fills a critical gap in our understanding of the impact of neonicotinoids on non-*Apis* pollinators. Bumblebees are known to be susceptible to pathogens and neonicotinoid exposures, but studies detailing interactions are very scarce. Other recent studies have described negative effects of neonicotinoids on bumblebee colony performance, reproductive production, and function, but none have examined effects in summer bumblebee colonies. The study also compliments nicely recent work describing both positive and neutral effects of neonicotinoids on immune functions in laboratory and small cage settings. The manuscript also relates to studies detailing effects of neonicotinoids on honey bee immunity and insect-pathogen relationships.

The study is also likely to be of considerable interest to researchers and people outside the immediate field. As a whole, bumblebees appear to be more seriously impacted by neonicotinoids at

field-relevant exposures than honey bees. Given the relative lack of impacts of neonicotinoids on honey bees at field-relevant exposures, policy decisions regarding neonicotinoid will likely be based on more susceptible pollinators. The fact that these are largely “negative results” describing a lack of interaction between clothianidin and bumblebees microbes does not detract from its value.

Data and Methodology: One of the strong points of this manuscript is the robust field design and thorough use of both colony performance metrics and microbe quantification. The authors not only performed a robust experiment, but their explanation of both the data and results is well ordered and easily interpreted. The authors present both significant and non-significant results (important for this type of experiment) in a transparent and thorough manner. I did not receive any extended data sets, but the authors could attach these in a data repository if needed.

Appropriate Use of Statistics and Treatment of Uncertainties: The statistical tests used here are appropriate for the design, and largely support the authors’ results. All of the error bars are defined in the figure headings.

One error that was noted was disagreement between the results text (line 162/163) and Supplementary Table 4. The p-values for the *G. apicola*-number of males interaction for the control and clothianidin groups differ between the text and the Table (control group 0.640 and 0.916; clothianidin group 0.020 and 0.010 respectively). However, these differences do not change the significance nor the appropriate conclusions for this finding.

Conclusions: The conclusions and data interpretations are robust, valid, and reliable given the robust experimental/field design. The only concern I have is that the virus prevalence in different colonies may be too low to provide an adequate challenge to the bumblebees’ pathogen resistance under the clothianidin stressor. A greater prevalence of the viruses among the colonies may have been a stronger test. None the less, there were no differences in pathogen abundances except for *C. bombi*.

Suggested Improvements: The manuscript could be improved by providing three points of additional information about effects of clothianidin on bumblebee colony performance.

- 1) Critically missing from this manuscript and Rundlöf et al., 2015 is a treatment comparison of the time it took colonies to produce a queen. The authors decided to end this study when the colonies produced their first queen rather than a fixed chronological time period. Given how critical development time is to production of reproductives and workers (two main effects impacted by clothianidin), a comparison of this metric would seem in order. If there is a treatment difference,

one group is rearing reproductives faster. If no difference exists, then clothianidin-treated colonies are less efficient at rearing reproductives (both in numbers and for males, size).

2) Also, the authors appear to have data that could solidly describe the negative effects of clothianidin exposure on worker brood rearing (number of worker cocoons). This metric is especially interesting since there was no apparent difference in adult worker numbers between the treatments. Comparison of the number of worker larvae successfully reared (as cocoons) is a rather direct measurement of colony brood rearing that gives better context to changes in reproductive-worker rearing ratios in bumblebee colonies.

3) Finally, Rundlöf et al., 2015 mentioned that they counted nectar/pollen cells but did not present it in their publication. This metric is a direct measurement of food storage/availability that informs narratives about relative forage efficiencies of treatment groups.

While not essential to these findings, all three metrics provide better context to interpret the observed differences in clothianidin-treated and untreated colony performances.

References: The authors cite previous studies appropriately throughout the manuscript. The subject of this study is well supported by references in related fields of bumblebee microbiology, neonicotinoid effects on bumblebee colony performance, and reproduction, and other bee pesticide-pathogen interactions.

Clarity and Context: The manuscript is very well written and well ordered in its presentation of the abstract, experimental design, description of methods and results, and discussion of the significance of the results. The one improvement that I would suggest is that the authors state more clearly which bees are specifically being referred to in parts of the introduction and discussion. In several places, the generic term “bee” is used when honey bee or bumblebee would be more illuminating. This change would highlight how little is known about bumblebee pesticide-pathogen interactions and other effects of neonicotinoids, and yet specify what is known. Of course, the authors should retain the “bees” designation when multiple bee species or bees in general are referred to.

Additional comments/corrections

line 160 ... and *G. apicola* and the number ...

line 162/163 Disagreement between p-values for the text and Supplementary Table 4 with *G. apicola* control and clothianidin (number of males)

line 201 Is there evidence for attraction of *Bombus terrestris* workers to neonicotinoid-contaminated food?

line 237 Do the authors mean "... did not affect virulence ..."?

Reviewer #2 (Remarks to the Author):

This is an interesting study expanding a previous one testing the effect of Clothianidin on bumblebees at field level.

The experimental plan is excellent, the methodology is sound and the conclusions are well supported. My opinion on the study is therefore very positive in general but I find the following point of weakness.

The authors clearly show that the negative effect previously observed by Rundlof et al, is related to a reduction in both the number and size of reproductives. Then, since no effects are noted on the prevalence and abundance of non-pathogenic microbes and intracellular parasites, authors conclude that bumblebee fitness is not impaired because of the possible effect of the treatment on pathogen susceptibility. Unfortunately, the authors do not further investigate the mechanistic link between the treatment and the biological effect they report.

The reason why the authors decided to test the possible correlation neonicotinoids-pathogens is clear in view of the already described possible interactions between neonicotinoids and parasites/pathogens. However, given the lack of correlation assessed in this study, one would expect that the authors explore the mechanisms underlying the effect they noted, since they clearly recognize that this is an important line of research (see lines 56 and 57).

For example, it is suggested that undernourishment of bumblebees in Clothianidin treated fields may play a relevant role and this is very likely, but if so, why the authors did not test this hypothesis by estimating, for example, the concentration of proteins/lipids/sugar in the haemolymph of bees under the different experimental conditions, to support their statement?

In other words, going back to the manuscript title: “Clothianidin field-exposure affects bumblebees directly rather than through increased pathogen susceptibility”, it seems to me that the authors concentrated too much on the second part of the sentence commencing with “rather than”, but forgot to deal with the adverb “directly” that comes before and looks like the most important conclusion.

Following are some more minor points.

Lines 112,113: are this data important?

Line 113: “male pupae”

Lines 116-118: cite Tab. 2?

Line 242: in principle, if a parasite is absent there is no way to induce a rise in its level, therefore, I would rather write “nor induce rise in the levels or virulence of intracellular parasites that were present”

Lines 261-263: maybe explain why the effect on nervous system should explain the synergism

Line 357: mL⁻¹, use exponent as in line 368, for example

Line 357 and following: normally one should write 65 °C with a space after the number, but please consult the formatting guide

Line 366: delete semicolon after “homogenization”

Line 386: insert a space after “10”

Statistical analysis: I’m not an expert but I wonder if some kind of correction for multiple comparisons should be applied when testing for a very high number of correlations as in Supplementary table 4

Line 466: maybe use “growth dynamics” instead of “growth characteristics”

References

There are a number of references where the scientific names are not in italic (e.g. 21,23,26,28 and many others) and others where they are (e.g. 64); please fix according to the formatting guide.

Ref. 46: “*Bombus*”

Ref. 44: "Plowright"

Ref. 61: "de Miranda"

Line 699: $P > 0.05$

Table 2, fifth column, sixth row "-0.90 mg"

Reviewer #3 (Remarks to the Author):

This paper reports important results that will be of wide interest. The writing is generally good, but there are areas in which the text needs substantial work to improve clarity and meaning. The authors have overlooked some relevant literature, and appear to be somewhat confused about what actually constitutes measures of fitness in bumblebees. The reference list contains many errors. I have outlined in comments to authors a number of suggested corrections/ edits and also flagged key areas that need more work.

L26-27: "We related clothianidin exposure and microbial composition to both individual- and colony-level fitness parameters, ..." Given that bumblebees are social insects, the unit of reproduction is the colony (rather than the individual), making measures of individual fitness parameters (as stated in this sentence) non-sensical. Presumably the authors mean individual measures of performance or similar. The current wording needs to be revised to avoid this inaccurate representation of reality.

L33: Is the term "synergism" here the most appropriate here? The interaction between clothianidin exposure and pathogens could be additive, rather than synergistic. Therefore I suggest the authors use "interactions", rather than "synergism", here.

L39-40: Consider citing some older evidence of these multiple interacting drivers of decline here (e.g. Vanbergen et al. 2013; Potts et al. 2009) alongside refs 1,2.

L50: the authors reference the study by Henry et al. (2012) to support this point, they should also cite the more recent work on homing success in honeybees (Fischer et al. 2014) and bumblebees (Stanley et al. 2016).

L50: The authors should also cite the study by Stanley et al. (2016), alongside refs 10 & 11, reporting that bumblebee foraging behaviour is affected by thiamethoxam exposure.

L48-51: Another major sublethal impact of neonicotinoid exposure, related to foraging behaviour, are impacts on the crop pollination services provided by bees. Impacts on pollination services have been reported in apple crops, where bumblebee colonies exposed to thiamethoxam visited flowers less often, collected less pollen and produced apples containing fewer seeds (Stanley et al. 2015). The authors should consider including this impact on ecosystem service provision by bees in response to neonicotinoid exposure.

L60-64: "Contrasting effects of neonicotinoid field-exposure on bees were in part ascribed act synergistically with pathogens in increasing bee mortality8,24,26-28." The authors have done a good job here of referencing studies reporting interactions between pesticide exposure and pathogen impacts, however there are a number of studies that have explicitly examined these questions and found no interaction (e.g. Baron et al. 2014; ref 43) that should also be cited here for completeness.

L72: The authors should cite Stanley et al. (2016) here alongside refs 10 and 11 relating to bumblebees collecting less pollen when exposed to neonicotinoids.

L78: I would question the use of the word "synergistic" again here. Studies set out to examine whether or not there are any interactions between exposure to neonicotinoids and pathogens. One type of interaction would be a synergism, but other interactions are also possible (e.g. no interaction, additive effects of combined exposure or even negative interactions such that exposure to both stressors results in a less severe impact than exposure to either stressor alone). I would strongly encourage the authors from cherry picking synergism as the outcome here.

L78-80: The authors should also cite ref 43 (Baron et al. 2017) here alongside refs 8, 31 and 41. Although there were no significant interactions between neonicotinoid exposure and pathogen exposure in this study (ref 43), it was designed to address this question specifically.

L88: "individual bee fitness parameters" - please change this wording (see my first comment, relating to the abstract) on this issue.

L89: suggest changing "potential synergism" to "potential interactive effects" here.

L91-92: "...exposure to clothianidin on bumblebee fitness by comparing the number of bees per caste, the body size of premature and adult bees, ..." The authors need to revise their wording here as these parameters are not measures of fitness (even at the colony level).

L94-96: "In addition, we test whether the co-variation between the amounts of microbiota and here presented and in Rundlöf et al. (2015)¹⁷ reported bumblebee fitness parameters was affected by clothianidin-exposure." This sentence needs revision as it currently does not make sense. What do the authors mean by "amounts of microbiota" - do they actually mean the structure of the microbial community? This sentence needs editing for content, grammar and meaning.

L100-102: Are these new results that are not previously reported in ref 17?

L106-110: It is unclear from this paragraph why the authors could not obtain more samples of male pupae from clothianidin-treated colonies, nor indeed why they could obtain so few samples of work pupae from control colonies. Do the authors conclude these differences are the result of random/stochastic processes outside the control of experimenters? Alternatively, do they represent a genuine treatment effect? The text here needs more information on what limited samples being collected here.

L143: suggest changing "Microorganism amounts" to "Microorganism abundance".

L150: Delete "with these".

L159: replace "amounts" with "abundance".

L164: replace "amounts" with "abundance".

L170: replace "amounts" with "abundance".

Results query: given that it was not possible to collect good data on the prevalence and abundance of microorganisms from all colonies, it is interesting to see a number of marginal p-values reported in the results section. I would encourage the authors to consider the scale and robustness of these analyses in terms of sample size/ replication. It would be very helpful to readers assessing the data

presented if they had some indication of the statistical power available to detect treatment differences.

L180: replace "reproductive" with "reproductive".

L180: remove citation of ref 16 here, which does not report on production of reproductive in honeybee colonies.

L181: replace "...colonies at clothianidin-treated fields compared to colonies at control fields" to read as follows - "...colonies in clothianidin-treated fields compared to colonies in control fields ..."

L198: The authors should cite Stanley et al. (2016) here alongside refs 10 and 11 relating to bumblebees collecting less pollen when exposed to neonicotinoids

L210: replace "amounts" with "abundance".

L218: the authors should cite ref 43 here (alongside ref 8).

L221: replace "amounts" with "abundance".

L243: add an apostrophe to "bumblebees".

L240-259: The authors should include reference here to work by Baron et al. (2014) examining the impacts of combined exposure to the pyrethroid (lambda cyhalothrin) and the pathogen *Crithidia bombii*. Their results showed that pesticide-treated colonies produced workers with a significantly lower body mass. However, Baron et al. (2014) reported that Lambda-cyhalothrin had no significant impact on the susceptibility of workers to *C. bombii*, or intensity of parasitic infection.

L453: replace "amounts" with "abundance".

L478 & 480: care with how you use the term "bee fitness" here - see earlier comments.

L488: replace "amounts" with "abundance".

L513: replace "A" with "a" after colon in article title for ref 3.

L520: remove extraneous upper case letters in article title (ref 6).

L543: use consistent abbreviated title for this journal - compare usage in ref 16 with ref 4.

L551: remove extraneous upper case letters in article title (ref 20).

L557: "*Bombus terrestris*" should be italics.

L559: replace "A" with "a" after colon in article title for ref 22.

L561: "*Zea mays*" should be italics.

L562: replace "1-20" with the article number for ref 23.

L564: replace "Sublethal" with "sublethal" after colon in article title for ref 24.

L568: "*Nosema*" should be italics.

L570-571: "*Nosema*" and "*Apis mellifera*" should be italics.

L573: Article number is missing from ref 27.

L574: replace "A" with "a" after colon in article title for ref 28.

L575: Article number is missing from ref 28.

L578: "*Apis mellifera*" should be italics.

L579: remove extraneous upper case letters in article title (ref 30).

L580: Article number is missing from ref 30.

L589: "*Aspergillus*" should be italics.

L591: replace "The" with "the" after colon in article title for ref 35.

L594: Are these the correct page numbers?

L595: remove extraneous upper case letters in article title (ref 37).

L602: "*Bombus*" should be italics.

L613: revise from "Plowriight," to "Plowright,"

L614: "*Bombus terricola*" should be italics.

L618: Capitalise "bombus", and this species name ("*Bombus terrestris*") should be in italics.

L623: remove extraneous upper case letters in article title (ref 48).

L624: "*Bombus terrestris*" should be in italics.

L626: "*Bombus terrestris*" should be in italics.

L630: replace "Pollinator" with "pollinator" after colon in article title for ref 51.

L634: "*Apis mellifera*" should be italics.

L639: "Chen, Y.-P. & Siede, R. in Advances in virus research (eds. Marmorosch, K., Shabalina, S. A. & Murphy, F.) 70, 33-80 (Elsevier Academic Press inc., 2007)." Check format of this reference - article title is missing. It is an edited annual journal, not a book chapter I believe.

L648: Article number is missing from ref 57.

L649: remove extraneous upper case letters in article title (ref 58).

L651: Article number is missing from ref 59.

L658: remove extraneous upper case letters in article title (ref 62).

L659: "*Apis mellifera*" should be italics.

L660: Article number is missing from ref 62.

L663: "*Bombus terrestris*" should be in italics.

Supplementary Info comments

Ref 67: "Nosema" and "Crithidia" should be in italics. Also, article number is missing from reference.

Ref 68: *Apis mellifera*" should be italics.

Ref 69: *Apis mellifera*" should be italics.

Supplementary Table 1: why has the non-significant (marginal) p-value of 0.057 been highlighted in this table?

References

Baron, G. L., N. E. Raine and M. J. F. Brown (2014). Impact of chronic exposure to a pyrethroid pesticide on bumblebees and interactions with a trypanosome parasite. *Journal of Applied Ecology* 51: 460-469.

Fischer, J., T. Muller, A.-K. Spatz, U. Greggers, B. Grunewald and R. Menzel. Neonicotinoids interfere with specific components of navigation in honeybees. *PLoS One* 9: e91364 (2014).

Potts, S. G. et al. Global pollinator declines: trends, impacts and drivers. *Trends in Ecology & Evolution* 25: 345-353 (2009).

Stanley, D. A. et al. Neonicotinoid pesticide exposure impairs crop pollination services provided by bumblebees. *Nature* 528: 548-550 (2015).

Stanley, D. A., A. L. Russell, S. J. Morrison, C. Rogers and N. E. Raine. Investigating the impacts of field-realistic exposure to a neonicotinoid pesticide on bumblebee foraging, homing ability and colony growth. *Journal of Applied Ecology* 53: 1440-1449 (2016).

Vanbergen, A. J. et al. Threats to an ecosystem service: pressures on pollinators. *Frontiers in Ecology and the Environment* 11: 251-259 (2013).

Point-by-point response to remarks by the editor and the reviewers:

Referee #	Remark from the Editor/Reviewers	Response from the Authors
	Referee #1	Response:
1	One error that was noted was disagreement between the results text (line 162/163) and Supplementary Table 4. The p-values for the G. apicola-number of males interaction for the control and clothianidin groups differ between the text and the Table (control group 0.640 and 0.916; clothianidin group 0.020 and 0.010 respectively). line 162/163 Disagreement between p-values for the text and Supplementary Table 4 with G. apicola control and clothianidin (number of males)	Corrected.
1	Critically missing from this manuscript and Rundlöf et al., 2015 is a treatment comparison of the time it took colonies to produce a queen. The authors decided to end this study when the colonies produced their first queen rather than a fixed chronological time period. Given how critical development time is to production of reproductives and workers (two main effects impacted by clothianidin), a comparison of this metric would seem in order. If there is a treatment difference, one group is rearing reproductives faster. If no difference exists, then clothianidin-treated colonies are less efficient at rearing reproductives (both in numbers and for males, size).	We agree that this is an interesting question. However, our experiment does not allow for the comparison of the time to produce queens as we freeze-killed colonies after the experiment was ended to count the number of cocoons, and adults of different castes and to analyse the microbiota of bumblebees. Clothianidin-exposed colonies had fewer premature and adult queens and males, which suggests that colonies switched later to the production of reproductives. We have now included a paragraph in the discussion on whether the reduction of reproductives represents a delay in or an abandonment of the production of reproductives: Lines 208-213: “When the colonies were terminated, the developmental stage of the worker

		brood was clearly more advanced than that of the male brood, suggesting that the colonies were switching from worker production to male/queen production. This suggests that the differential between exposed and control colonies in the preponderance of worker or male brood may represent a delay in, rather than an abandonment of, the production of reproductives.” We also added a sentence acknowledging some uncertainty in the counts of adult workers (Lines 202-205) and an accompanying sentence to the methods to make this comprehensible (Lines 350-353). We also added a sentence on the bee categorization (Lines 365-366).
1	Also, the authors appear to have data that could solidly describe the negative effects of clothianidin exposure on worker brood rearing (number of worker cocoons). This metric is especially interesting since there was no apparent difference in adult worker numbers between the treatments. Comparison of the number of worker larvae successfully reared (as cocoons) is a rather direct measurement of colony brood rearing that gives better context to changes in reproductive-worker rearing ratios in bumblebee colonies.	This is an interesting question, which can, however, not be directly addressed with our dataset. We only counted premature and adult bumblebees after termination of the experiment. In addition, counts of pupae per caste were not exhaustive, as we simply aimed at obtaining samples of ten pupae of a caste (or of both castes (workers and males), in colonies in which both castes were present in the small cocoons). Our findings that there was no statistically significant treatment difference in small cocoons (i.e. either workers or males) or in adult workers suggests that there are no major differences in worker brood rearing. We can however not exclude the possibility that colonies reared more workers relatively late in the season to compensate for lower production rates or higher mortality rates earlier in the season.
1	Finally, Rundlöf et al., 2015 mentioned that they counted	We have now included these metrics in the discussion and refer

	nectar/pollen cells but did not present it in their publication. This metric is a direct measurement of food storage/availability that informs narratives about relative forage efficiencies of treatment groups.	to the analysis of a possible effect of clothianidin exposure on the number of nectar and pollen cells presented in Extended Data Table 5 of Rundlöf et al. (2015).
1	The one improvement that I would suggest is that the authors state more clearly which bees are specifically being referred to in parts of the introduction and discussion. In several places, the generic term “bee” is used when honey bee or bumblebee would be more illuminating. This change would highlight how little is known about bumblebee pesticide-pathogen interactions and other effects of neonicotinoids, and yet specify what is known. Of course, the authors should retain the “bees” designation when multiple bee species or bees in general are referred to.	As suggested by the reviewer, we have specified where relevant the type/species of bee throughout the manuscript, for example: Lines 65-67: “Neonicotinoid exposure was shown to increase pathogen abundance in honeybees ^{14,25-27} but not in bumblebees ^{8,33} and to act synergistically with pathogens in increasing mortality of honeybees ^{26,28,29} and bumblebees ⁸ ” Lines 223-225: “ Bombus terrestris and honeybee workers can be attracted to neonicotinoid contaminated food, even though its consumption can reduce the bees’ overall food intake ⁵⁰ .”
	line 160 ... and G. apicola and the number ...	We rephrased this sentence for linguistic reasons: Lines 172-174: Clothianidin exposure affected how G. apicola abundance in adult workers co-varied with the body mass of adult workers (P=0.006) and the number of adult males (P=0.027, Supplementary Table 4)
1	line 201 Is there evidence for attraction of Bombus terrestris workers to neonicotinoid-contaminated food?	Yes. We have specified this now.
1	line 237 Do the authors mean “... did not affect virulence ...”?	Yes. This has been corrected.
	Referee #2	Response:
2	However, given the lack of correlation assessed in this study, one would expect that the authors explore the mechanisms	We agree that this is an interesting aspect worth studying in more detail. Unfortunately this was not feasible within the scope of this

	underlying the effect they noted, since they clearly recognize that this is an important line of research (see lines 56 and 57). For example, it is suggested that undernourishment of bumblebees in Clothianidin treated fields may play a relevant role and this is very likely, but if so, why the authors did not test this hypothesis by estimating, for example, the concentration of proteins/lipids/sugar in the haemolymph of bees under the different experimental conditions, to support their statement? In other words, going back to the manuscript title: “Clothianidin field-exposure affects bumblebees directly rather than through increased pathogen susceptibility”, it seems to me that the authors concentrated too much on the second part of the sentence commencing with “rather than”, but forgot to deal with the adverb “directly” that comes before and looks like the most important conclusion.	study. Such mechanistic studies are furthermore better realized with a more precise, dedicated experimental design. In Rundlöf et al. (2015) we analysed the number of pollen and nectar cells, which can serve as a proxy for a colony’s ability to collect food. We have now included in the discussion that we found no treatment effect on these matrices and we discuss the possibility that the foraging ability of bumblebees may be affected by neonicotinoid exposure through the published literature. In recognition of the reviewer’s valid criticism, we have changed the title to “Clothianidin field exposure affects bumblebees but generally not their pathogens” thereby avoiding undue emphasis in the title on the possible mechanisms of these effects.
2	Lines 112,113: are this data important?	We believe that these data are relevant because (i) apparent sex-differences in body mass make it necessary to differentiate between sexes in the subsequent analyses and (ii) the more advanced developmental stage of workers relative to males suggests that the colonies with workers present were at a point where they had just switched, or were about to switch, to the production of males instead of workers. We have now added the second point to the discussion.
2	Line 113: “male pupae”	Revised as suggested.

2	Lines 116-118: cite Tab. 2?	Revised as suggested.
2	Line 242: in principle, if a parasite is absent there is no way to induce a rise in its level, therefore, I would rather write “nor induce rise in the levels or virulence of intracellular parasites that were present”	Revised as suggested.
2	Lines 261-263: maybe explain why the effect on nervous system should explain the synergism	Since bee viruses and neonicotinoids share a common target – the bee nervous system – it is logically plausible that there could be synergism between neonicotinoids and viruses in their effects on bees. We rephrased the relevant sentence to clarify this argument.
2	Line 357: mL ⁻¹ , use exponent as in line 368, for example	Revised as suggested.
2	Line 357 and following: normally one should write 65 °C with a space after the number, but please consult the formatting guide Line 386: insert a space after “10”	Revised as suggested.
2	Line 366: delete semicolon after “homogenization”	Revised as suggested.
2	Statistical analysis: I’m not an expert but I wonder if some kind of correction for multiple comparisons should be applied when testing for a very high number of correlations as in Supplementary table 4	We are aware of the risk of obtaining false positive (Type-I) statistical errors when testing for a large number of parameters, and guard against this conservatively in our interpretation of the data and its conclusions. However, we do not assume that each parasite affects the bee performance parameters equally, i.e. the hypotheses differ depending on the microorganism and the bee performance parameters as the microorganisms differ in their modes of action. Therefore we believe post-hoc analyses are not necessarily required. The only statistically significant differences

		in Supplementary Table 4 that may turn non-significant after applying a post-hoc correction are the ones concerning the number of adult males. We have now included in the discussion the possibility that these items may reflect Type-I errors, rather than biological effects.
2	Line 466: maybe use “growth dynamics” instead of “growth characteristics”	Revised as suggested.
2	References There are a number of references where the scientific names are not in italic (e.g. 21,23,26,28 and many others) and others where they are (e.g. 64); please fix according to the formatting guide. Ref. 46: “Bombus” Ref. 44: “Plowright” Ref. 61: “de Miranda”	We corrected the misspelled author names and wrote scientific names in titles in italics.
2	Line 699: P>0.05	Corrected.
2	Table 2, fifth column, sixth row “-0.90 mg”	We transformed sqrt-mg back to mg.
	Referee #3	Response:
3	L26-27: "We related clothianidin exposure and microbial composition to both individual- and colony-level fitness parameters, ..." Given that bumblebees are social insects, the unit of reproduction is the colony (rather than the individual), making measures of individual fitness parameters (as stated in this sentence) non-sensical. Presumably the authors mean individual measures of performance or similar. The current wording needs to be revised to avoid this inaccurate representation of reality.	We replaced the term “(bee) fitness” with “(bee) performance” throughout the manuscript.

	L88: "individual bee fitness parameters" - please change this wording (see my first comment, relating to the abstract) on this issue. L91-92: "...exposure to clothianidin on bumblebee fitness by comparing the number of bees per caste, the body size of premature and adult bees, ..." The authors need to revise their wording here as these parameters are not measures of fitness (even at the colony level). L478 & 480: care with how you use the term "bee fitness" here - see earlier comments.	
3	L33: Is the term "synergism" here the most appropriate here? The interaction between clothianidin exposure and pathogens could be additive, rather than synergistic. Therefore I suggest the authors use "interactions", rather than "synergism", here. L78: I would question the use of the word "synergistic" again here. Studies set out to examine whether or not there are any interactions between exposure to neonicotinoids and pathogens. One type of interaction would be a synergism, but other interactions are also possible (e.g. no interaction, additive effects of combined exposure or even negative interactions such that exposure to both stressors results in a less severe impact than exposure to either stressor alone). I would strongly	Revised as suggested.

	encourage the authors from cherry picking synergism as the outcome here. L89: suggest changing "potential synergism" to "potential interactive effects" here.	
3	L39-40: Consider citing some older evidence of these multiple interacting drivers of decline here (e.g. Vanbergen et al. 2013; Potts et al. 2009) alongside refs 1,2.	We are conscious of our indebtedness to our predecessors, and the published record. However, in order to comply with the journal's reference limit, we occasionally had to make hard choices, and in such cases the more recent literature represents a more complete record of historical achievement than older literature. We would naturally prefer a more inclusive recognition of the published record, through additional references, if the editors will allow this.
3	L50: the authors reference the study by Henry et al. (2012) to support this point, they should also cite the more recent work on homing success in honeybees (Fischer et al. 2014) and bumblebees (Stanley et al. 2016). L50: The authors should also cite the study by Stanley et al. (2016), alongside refs 10 & 11, reporting that bumblebee foraging behaviour is affected by thiamethoxam exposure.	We have now included the suggested literature.
3	L48-51: Another major sublethal impact of neonicotinoid exposure, related to foraging behaviour, are impacts on the crop pollination services provided by bees. Impacts on pollination services have been reported in apple crops, where bumblebee colonies exposed to thiamethoxam visited flowers less often, collected less pollen and produced apples containing	Revised as suggested.

	fewer seeds (Stanley et al. 2015). The authors should consider including this impact on ecosystem service provision by bees in response to neonicotinoid exposure.	
3	L60-64: "Contrasting effects of neonicotinoid field-exposure on bees were in part ascribed act synergistically with pathogens in increasing bee mortality8,24,26-28." The authors have done a good job here of referencing studies reporting interactions between pesticide exposure and pathogen impacts, however there are a number of studies that have explicitly examined these questions and found no interaction (e.g. Baron et al. 2014; ref 43) that should also be cited here for completeness.	We have now cited Baron et al. (2017) here, but not Baron et al. (2014), since our focus is on neonicotinoid-pathogen interactions and the experiment in Baron et al. (2014) focused on a pyrethroid rather than a neonicotinoid. However, we cite Baron et al. (2014) in the discussion, where our study is discussed in the wider context of pesticide-pathogen interactions.
3	L72: The authors should cite Stanley et al. (2016) here alongside refs 10 and 11 relating to bumblebees collecting less pollen when exposed to neonicotinoids.	Revised as suggested.
3	L78-80: The authors should also cite ref 43 (Baron et al. 2017) here alongside refs 8, 31 and 41. Although there were no significant interactions between neonicotinoid exposure and pathogen exposure in this study (ref 43), it was designed to address this question specifically.	Revised as suggested.
3	L100-102: Are these new results that are not previously reported in ref 17?	Yes, these are new results. We rephrased the final paragraphs of the introduction to state more precisely what is the new data analysed in this study.
3	L106-110: It is unclear from this paragraph why the authors could not obtain more samples of male	We revised this paragraph and added biological context, by emphasizing that these

	pupae from clothianidin-treated colonies, nor indeed why they could obtain so few samples of work pupae from control colonies. Do the authors conclude these differences are the result of random/ stochastic processes outside the control of experimenters? Alternatively, do they represent a genuine treatment effect? The text here needs more information on what limited samples being collected here.	observations and sample distributions are most likely a consequence of delayed colony development in the clothianidin-exposed colonies, relative to non-exposed colonies. Lines 119-121: "Because the control colonies tended to be further in their development than the exposed colonies, we were able to obtain male pupal samples from 28 of 32 colonies at untreated fields, but from only 16 of 32 colonies at clothianidin-treated fields. Similarly, we were able to obtain samples of at least 7 worker pupae more often from clothianidin-exposed (18) than control colonies (4). Samples of both male and worker pupae could be obtained from four clothianidin-exposed colonies that were in the transition from worker to male production, while two exposed colonies had neither worker nor male pupae.
3	L94-96: "In addition, we test whether the co-variation between the amounts of microbiota and here presented and in Rundlöf et al. (2015)¹⁷ reported bumblebee fitness parameters was affected by clothianidin-exposure." This sentence needs revision as it currently does not make sense. What do the authors mean by "amounts of microbiota" - do they actually mean the structure of the microbial community? This sentence needs editing for content, grammar and meaning. L143: suggest changing "Microorganism amounts" to "Microorganism abundance".	As suggested, we replaced the term "(microorganism) amounts" by "(microorganism) abundance". We also rephrased the final paragraph of the introduction and removed the confusing sentence at the end. .

	L159: replace "amounts" with "abundance". L164: replace "amounts" with "abundance". L170: replace "amounts" with "abundance". L210: replace "amounts" with "abundance". L221: replace "amounts" with "abundance". L453: replace "amounts" with "abundance". L488: replace "amounts" with "abundance".	
3	L150: Delete "with these".	Revised as suggested.
3	Results query: given that it was not possible to collect good data on the prevalence and abundance of microorganisms from all colonies, it is interesting to see a number of marginal p-values reported in the results section. I would encourage the authors to consider the scale and robustness of these analyses in terms of sample size/ replication. It would be very helpful to readers assessing the data presented if they had some indication of the statistical power available to detect treatment differences.	We added a Supplementary Figure showing statistical power for treatment effects and interactive effects between treatment and microorganism abundance where $0.05 < P < 0.1$.
3	L180: replace "reproductive" with "reproductives".	We replaced "reproductive" with "reproductives" (plural)
3	L180: remove citation of ref 16 here, which does not report on production of reproductive in	Revised as suggested.

	honeybee colonies.	
3	L181: replace "...colonies at clothianidin-treated fields compared to colonies at control fields" to read as follows - "...colonies in clothianidin-treated fields compared to colonies in control fields ..."	The colonies were actually not placed in(side) the fields but rather beside the fields. We have clarified this in all major sections and changed the sentence in question to: "...colonies next to clothianidin-treated fields compared to colonies next to control fields ..." For simplicity, we have, however, kept the preposition "at" in some parts of the manuscript, where the focus is on results rather than the study design.
3	L198: The authors should cite Stanley et al. (2016) here alongside refs 10 and 11 relating to bumblebees collecting less pollen when exposed to neonicotinoids	Revised as suggested.
3	L243: add an apostrophe to "bumblebees".	Revised as suggested.
3	L240-259: The authors should include reference here to work by Baron et al. (2014) examining the impacts of combined exposure to the pyrethroid (lambda cyhalothrin) and the pathogen Crithidia bombii . Their results showed that pesticide-treated colonies produced workers with a significantly lower body mass. However, Baron et al. (2014) reported that Lambda-cyhalothrin had no significant impact on the susceptibility of workers to C. bombi , or intensity of parasitic infection.	We have included this study in the discussion: Lines 300-304: "Pathogen-pesticide interaction in bumblebees has been studied under laboratory conditions with the pyrethroid λ-cyhalothrin and C. bombi⁶¹. Chronic exposure to the pyrethroid did not affect C. bombi prevalence or abundance but the body mass of B. terrestris workers. Other individual-level or colony-level performance parameters were unaffected by the treatment⁶¹."
3	L513: replace "A" with "a" after colon in article title for ref 3. L520: remove extraneous upper	We corrected the indicated errors in the reference list.

case letters in article title (ref 6). L543: use consistent abbreviated title for this journal - compare usage in ref 16 with ref 4. L551: remove extraneous upper case letters in article title (ref 20). L557: "Bombus terrestris" should be italics. L559: replace "A" with "a" after colon in article title for ref 22. L561: "Zea mays" should be italics. L562: replace "1-20" with the article number for ref 23. L564: replace "Sublethal" with "sublethal" after colon in article title for ref 24. L568: "Nosema" should be italics. L570-571: "Nosema" and "Apis mellifera" should be italics. L573: Article number is missing from ref 27. L574: replace "A" with "a" after colon in article title for ref 28. L575: Article number is missing from ref 28. L578: "Apis mellifera" should be italics. L579: remove extraneous upper case letters in article title (ref 30). L580: Article number is missing	
---	--

from ref 30. L589: "Aspergillus" should be italics. L591: replace "The" with "the" after colon in article title for ref 35. L594: Are these the correct page numbers? L595: remove extraneous upper case letters in article title (ref 37). L602: "Bombus" should be italics. L613: revise from "Plowriight," to "Plowright," L614: "Bombus terricola" should be italics. L618: Captialise "bombus", and this species name ("Bombus terrestris") should be in italics. L623: remove extraneous upper case letters in article title (ref 48). L624: "Bombus terrestris" should be in italics. L626: "Bombus terrestris" should be in italics. L630: replace "Pollinator" with "pollinator" after colon in article title for ref 51. L634: "Apis mellifera" should be italics. L639: "Chen, Y.-P. & Siede, R. in Advances in virus research (eds. Marmorosch, K., Shabalina, S. A. & Murphy, F.)	
---	--

	70, 33-80 (Elsevier Academic Press inc., 2007)." Check format of this reference - article title is missing. It is an edited annual journal, not a book chapter I believe. L648: Article number is missing from ref 57. L649: remove extraneous upper case letters in article title (ref 58). L651: Article number is missing from ref 59. L658: remove extraneous upper case letters in article title (ref 62). L659: "Apis mellifera" should be italics. L660: Article number is missing from ref 62. L663: "Bombus terrestris" should be in italics. Supplementary Info comments Ref 67: "Nosema" and "Crithidia" should be in italics. Also, article number is missing from reference. Ref 68: "Apis mellifera" should be italics. Ref 69: "Apis mellifera" should be italics.	
3	Supplementary Table 1: why has the non-significant (marginal) p-value of 0.057 been highlighted in this table?	We removed the highlight.

REVIEWERS' COMMENTS:

Reviewer #1 (Remarks to the Author):

This revised manuscript by Wintermantel et al. is considerably improved over the original. The authors addressed the concerns of the reviewers. Most corrections and additions were points of clarification and section delineating the scope and interpretation of the results. Relatively modest changes in terminology corrected some problems with the manuscript and provided better context for the results. In particular, several key findings were rephrased to better capture what was examined (i.e. bumblebee performance, interactions). Parts of the manuscript that originally appeared as shortfalls or gaps in the study were adequately explained and given better context. Most important, the revised manuscript better explains how changes in individual performance could affect colony fitness (production of reproductives), which is a major effect well worth reporting. The study now reports both the deleterious effects of clothianidin on bumblebee reproductives and the general lack of effects on microbiota with equal emphasis. The authors also related their findings to the literature based in part on references suggested by another reviewer.

I have relatively few corrections and suggestions to make, given the improved state of the new manuscript.

- 1) I haven't been able to see the raw data that might be presented upon request. However, the results are clearly arranged and well presented.
- 2) The authors should check the references to make sure that species are italicized throughout.

Reviewer #3 (Remarks to the Author):

This paper reports important results that will be of wide interest and makes a valuable contribution to the field. The authors have made a thorough revision of the manuscript and I am very happy that they have addressed all the comments and criticisms I raised to my satisfaction.

I was also asked by the editorial office to offer my comments on how the authors took the comments of reviewer 2 into account. In my opinion the authors have done a very thorough job of

responding to the few comments raised by reviewer 2, and made all the changes to their manuscript necessary to deal with the issues raised.

In the process of revising the manuscript a few very minor issues have crept in that need to be fixed before this is accepted - see short list below (perhaps 5-10 minutes of work to make these minor corrections).

L24: change "analyse" to "analysed".

L26: change "relate" to "related".

L29: change "find" to "found".

L30: change "reduces" to "reduced".

L30: change "does" to "did".

L119: "Because the control colonies tended to be further in their development..." should be changed to "Because the control colonies tended to be further along in their development..."

L208: Ref 10 should be cited here alongside 11 and 12.

L658: "Jansen, V. A. A. A., Brown, M. J. F. F." should be changed to "Jansen, V. A. A., Brown, M. J. F." (for some reason the authors third initial has been duplicated in both cases.

L691: "Bombus" should be in italics.

Point-by-point response to remarks by and the reviewers:

Referee #	Remark from the Reviewers	Response from the Authors
	Referee #1	Response:
1	I haven't been able to see the raw data that might be presented upon request. However, the results are clearly arranged and well presented.	The raw data are available upon request. Besides, the Figures show not only model estimates and confidence intervals but also individual measurements.
1	The authors should check the references to make sure that species are italicized throughout.	We have corrected this now.
	Referee #3	Response:
3	L24: change "analyse" to "analysed".	Revised as suggested
3	L26: change "relate" to "related".	Revised as suggested
3	L29: change "find" to "found".	We have now replaced "find" by "show" as Nature communications' formatting checklist proposes to start the presentation of the major results with " 'Here we show' or similar".
3	L30: change "reduces" to "reduced".	The journal's formatting checklist states that in the abstract "Results of the current study are written in present tense ". We are happy with either tense here and leave the decision to use present or past tense with the editor.
3	L30: change "does" to "did".	See response above.
3	L119: "Because the control colonies tended to be further in their development..." should be changed to "Because the control colonies tended to be further along in their development..."	Revised as suggested
3	L208: Ref 10 should be cited here alongside 11 and 12.	Revised as suggested
3	L658: "Jansen, V. A. A. A., Brown, M. J. F. F." should be changed to	We have corrected this now.

	"Jansen, V. A. A., Brown, M. J. F." (for some reason the authors third initial has been duplicated in both cases.	
3	L691: "Bombus" should be in italics.	We have corrected this now.